# Maf/ham1-like pyrophosphatases of non-canonical nucleotides are host-specific partners of viral RNA-dependent RNA polymerases

**Adrian A. Valli**[1]*, **Rafael García López**[1], **María Ribaya**[1¤a], **Francisco Javier Martínez**[1], **Diego García Gómez**[2], **Beatriz García**[1], **Irene Gonzalo**[1], **Alfonso Gonzalez de Prádena**[1], **Fabio Pasin**[1¤b], **Inmaculada Montanuy**[3], **Encarnación Rodríguez-Gonzalo**[2], **Juan Antonio García**[1]

**1** Centro Nacional de Biotecnología (CNB-CSIC), Madrid, Spain, **2** Departamento de Química Analítica, Nutrición y Bromatología, Universidad de Salamanca, Salamanca, Spain, **3** Facultad de Ciencias Experimentales, Universidad Francisco de Vitoria, Madrid, Spain

¤a Current address: Centre for Research in Agricultural Genomics (CRAG-CSIC-IRTA-UAB-UB), Cerdanyola del Vallès, Barcelona, Spain
¤b Current address: Instituto de Biología Molecular y Celular de Plantas (IBMCP, UPV-CSIC), Valencia, Spain
* avalli@cnb.csic.es

**Data Availability Statement:** All relevant data are within the manuscript and its Supporting Information files.

## Abstract

Cassava brown streak disease (CBSD), dubbed the "Ebola of plants", is a serious threat to food security in Africa caused by two viruses of the family *Potyviridae*: cassava brown streak virus (CBSV) and Ugandan (U)CBSV. Intriguingly, U/CBSV, along with another member of this family and one secoviridae, are the only known RNA viruses encoding a protein of the Maf/ham1-like family, a group of widespread pyrophosphatase of non-canonical nucleotides (ITPase) expressed by all living organisms. Despite the socio-economic impact of CDSD, the relevance and role of this atypical viral factor has not been yet established. Here, using an infectious cDNA clone and reverse genetics, we demonstrate that UCBSV requires the ITPase activity for infectivity in cassava, but not in the model plant *Nicotiana benthamiana*. HPLC-MS/MS experiments showed that, quite likely, this host-specific constraint is due to an unexpected high concentration of non-canonical nucleotides in cassava. Finally, protein analyses and experimental evolution of mutant viruses indicated that keeping a fraction of the yielded UCBSV ITPase covalently bound to the viral RNA-dependent RNA polymerase (RdRP) optimizes viral fitness, and this seems to be a feature shared by the other members of the *Potyviridae* family expressing Maf/ham1-like proteins. All in all, our work (i) reveals that the over-accumulation of non-canonical nucleotides in the host might have a key role in antiviral defense, and (ii) provides the first example of an RdRP-ITPase partnership, reinforcing the idea that RNA viruses are incredibly versatile at adaptation to different host setups.

**Funding:** This work was supported by grants BIO2015-73900-JIN (A.A.V) and BIO2016-80572-R (J.A.G) funded by MCIN/ AEI /10.13039/501100011033/ and by FEDER a way to make Europe; grants PID2019-110979RB-I00 (A.A.V) and PID2019-109380RBI00 (J.A.G) funded by MCIN/ AEI /10.13039/501100011033; and grant RYC2018-025523-I (A.A.V) funded by MCIN/ AEI /10.13039/501100011033 and El FSE invest in your future. The funders had no role in study design, data collection and analysis, decision to publish, or preparation of the manuscript.

**Competing interests:** The authors declare that they have no conflict of interest.

## Author summary

Cassava is one the most important staple food around the world in term of caloric intake. The cassava brown streak disease, caused by cassava brown streak virus (CBSV) and Ugandan (U)CBSV –*Ipomovirus* genus, *Potyviridae* family-, produces massive losses in cassava production. Curiously, these two viruses, unlike the vast majority of members of the family, encode a Maf1/ham1-like pyrophosphatase (HAM1) of non-canonical nucleotides with unknown relevance and function in viruses. This study aims to fill this gap in our knowledge by using reverse genetics, biochemistry, metabolomics and directed virus evolution. Hence, we found that HAM1 is required for UCBSV to infect cassava, where its pyrophosphatase activity resulted critical, but not to propagate in the model plant *Nicotiana benthamiana*. In addition, we demonstrated that HAM1 works in partnership with the viral RdRP during infection. Unexpected high levels of ITP/XTP non-canonical nucleotides found in cassava, and the known flexibility of RNA viruses to incorporate additional factors when required, supports the idea that the high concentration of ITP/XTP worked as a selection pressure to promote the acquisition of HAM1 into the virus in order to promote a successful infection.

## Introduction

The family *Potyviridae* is the largest and most socio-economically relevant group of plant-infecting RNA viruses. With more than 200 assigned members sorted in 12 different genera, these viruses represent a major threat for basically every important crop on earth. Potyvirids (members of the family *Potyviridae*) share common features, such as (i) monopartite (except for a few bipartite viruses) and positive sense single-stranded RNA (+ssRNA) genome, (ii) transmission mediated by vectors, and (iii) picorna-like gene expression strategy based on large polyproteins further processed by viral-encoded proteinases [1–3]. Potyvirids, in most cases, produce 10 mature proteins: P1, HCPro, P3, P3N-PIPO, 6K1, CI, 6K2, NIa (VPg/NIa-Pro), NIb and CP. Of relevance to this study, NIaPro is a *cis-* and *trans*-acting proteinase that releases most of the mature factors from the polyprotein [4,5], and NIb is a RNA-dependent RNA polymerase (RdRP) that replicates the viral genome [6,7].

With seven members described so far, the *Ipomovirus* genus is the most versatile group of potyvirids in term of genome organization, since only two of them follow the most common arrangement mentioned above [8]. The remaining five ipomoviruses lack the HCPro coding region and express either one P1 proteinase or two P1s in tandem [9–11]. Two viruses infecting *Manihot esculenta* (cassava) in nature are classified in this genus: cassava brown streak virus (CBSV) and Ugandan cassava brown streak virus (UCBSV), which cause the devastating cassava brown streak disease (CBSD), also dubbed the "Ebola of plants" [12,13]. Indeed, CBSD is considered among the seven most detrimental plant diseases in the world for its impact on the economy and food security in Africa, where it causes about 750 million US$ annual losses just in Tanzania, Uganda, Kenya and Malawi [14,15].

Even though CBSV and UCBSV are two distinct viral species, their genomes share around 72% nucleotide sequence identity, just below the species demarcation criteria in potyvirids (76%) [16]. Moreover, these two viruses (i) encode a single P1 leader proteinase, (ii) lack HCPro and, as the most striking feature, (iii) present an extra cistron between NIb and CP that encodes a *bona fide* Maf/Ham1-like protein [11]. This protein (referred as HAM1 in this study) belongs to the inosine triphosphate (ITP) pyrophosphatase (ITPase) family, which hydrolyzes the pyrophosphate bonds in triphosphate substrates (ITP/XTP) to release the

corresponding monophosphate (IMP/XMP) and a pyrophosphate molecule [17–20]. The presence of putative cleavage sites for the NIaPro proteinase at the N- and C-termini of HAM1 suggested that this protein accumulates into infected cells as a free product [11].

HAM1-like enzymes are present in prokaryotes and eukaryotes, across all life kingdoms, where they are proposed to prevent (i) incorporation of non-canonical nucleotides into nascent DNA and RNA molecules, (ii) RNA mistranslation, and (iii) inhibition of ATP-dependent enzymes [21]. Although they are widespread in nature, HAM1-like proteins are not usually encoded in viral genomes; in fact, their presence has been reported in only four RNA viruses so far. Intriguingly, all these HAM1-expressing RNA viruses infect plants from the *Euphorbiaceae* family: three potyvirids [CBSV, UCSBV and euphorbia ringspot virus (EuRV, *Potyvirus* genus)] [11,22], and one virus from the *Secoviridae* family [cassava torrado-like virus (CsTLV)] [23]. Even though a recent study has shown that CBSV and UCBS HAM1s are genuine pyrophosphatases in *in vitro* experiments, and that they determine necrotic symptoms in the model plant *Nicotiana benthamiana* [24], relevance and defined role of viral-derived HAM1 proteins are still unknown.

In this study, among other approaches, we used reverse genetics to manipulate an infectious cDNA clone of UCBSV in order to gain insight about the role of RNA virus-derived HAM1 proteins. Briefly, our experiments revealed that: (i) HAM1 is required for the virus to infect cassava, but not to produce a successful infection in the model plant *N. benthamiana*, and (ii) it works in partnership with the viral RdRP. The extremely high levels of non-canonical nucleotides that we have found in cassava, and likely present in other *Euphorbiaceae* plants, should have worked as a strong selection pressure to promote the acquisition of an ITP/XTP pyrophosphatase activity into virus RdRP in order to support successful replication and infection.

## Materials and methods

### Plants

Cassava plants (cultivar 60444) were grown in a chamber with 16h/8h light/dark cycles at 28˚C. *N. benthamiana* plants were grown in a greenhouse with 16h/8h light/dark cycles at 20-to-24˚C with supplementary light. For viral infection, *N. benthamiana* plants were moved just after inoculation to the cassava-growing chamber.

### Plasmids

Oligonucleotides used for this study are listed in S1 Table. UCBSV full-length clones derive from pLX-UCBSVi, a version of pLX-UCBSV (GenBank KY825157.1) [25] that carries the second intron of *Solanum tuberosum* ST-LS1 gene to interrupt the UCBSV P3 cistron. To generate pLX-UCBSVi, the mentioned intron was first amplified by PCR from pIC-PPV [26] with primers #3257/#3258. The 3'-half part of the UCBSV P3 cistron was amplified by PCR with primers #3259/#3260. An overlapping PCR with primers #3257/#3260 was used to join these two PCR products [intron-P3(3´half)]. UCBSV P1 and the 5'-half part of the UCBSV P3 cistron [P1-P3(5'-half)] were amplified with primers #3255/#3256. Finally, a DNA fragment that carries P1-P3(5'-half)-intron-P3(3´half) was produced by overlapping PCR with primers #3255/#3260, using P1-P3(5'-half) and intron-P3(3´half) as templates. This PCR product was digested with Bsu36I and NheI and introduced by ligation in pLX-UCBSV, which had been digested with the same enzymes, to replace the equivalent intron-less DNA segment.

To generate pLX-UCBSVi-eGFP (a GFP-tagged version of UCBSV), pLX-UCBSVi was used as backbone to introduce the GFP coding sequence between the HAM1 and CP cistrons. To allow the release of GFP during the infection, its coding sequence was flanked at both sides by synonymous sequences encoding the NIaPro cleavage site located between HAM1 and CP

(LTIDVQ/A). First, eGFP (F64L, S65A, V163A) coding sequence was amplified by PCR with primers #3360/#3361, adding the coding sequence of NIaPro cleavage site in the reverse primer, by using P1P1b clone [27] as template. Then, the N-terminus of NIb and the whole HAM1 coding sequences were amplified by PCR with primers #3160/#3358, adding the coding sequence of the NIaPro cleavage site in the reverse primer, by using pLX-UCBSVi as template. A subsequent overlapping PCR with primers #3160//#3361 was used to join the two above-mentioned PCR products into one single DNA segment. Finally, a BstBI/StuI fragment (the last 25 nt from NIb, the whole HAM1 and 1 nt from CP) from pLX-UCBSVi was replaced by the larger PCR product digested with BstBI.

To generate a 2xMyc-tagged version of HAM1 in UCSBV, pLX-UCBSVi was used as backbone to introduce the 2xMYC (GLINGEQKLISEEDLNGEQKLISEEDL) coding sequence just upstream the coding sequence that corresponds to the NIaPro cleavage site located between HAM1 and CP. First, the N-terminus of NIb and most of HAM1 coding sequences were PCR amplified with primers #3160/#3162, adding the coding sequence of 1xMyc in the reverse primer, by using pLX-UCBSVi as template. Then, a second PCR with primers #3160/#3163, adding the coding sequence of another 1xMyc and the NIaPro cleavage site (LTIDVQ/) in the reverse primer, was carried out by using the first PCR product as template. Finally, a BstBI/StuI fragment (the last 25 nt from NIb, the whole HAM1 and 1 nt from CP) from pLX-UCBSVi was replaced by the second PCR product digested with BstBI/StuI to generate pLX-UCBSVi-2xMyc.

Mutagenesis of HAM1 in both pLX-UCBSVi and pLX-UCBSVi-HAM1-2xMyc backbones was done by using a previously described method [28]. In brief, two PCR products having overlapping ends, which carry the desired mutation, were used as template of a subsequent PCR to join both PCR products in a larger DNA fragment. Then, a BstBI/StuI fragment (the last 25 nt from NIb, the whole HAM1 and 1 nt from CP) in the corresponding backbone was replaced by the indicated PCR products digested with the same enzymes. A list of pLX-UCBSVi- and pLX-UCBSVi-HAM1-2xMyc-derivatives, as well as the name of primers used for the amplification of different inserts, are shown in S2 Table.

The plasmid that expresses UCBSV-HAM1$_{T1A/D3G}$-2xMyc, a double mutant that carries T1A and D3G mutations in HAM1, was generated by replacing the BstBI/StuI fragment in pLX-UCBSVi-HAM1-2xMyc with the RT-PCR product amplified with primers #3160/#3130 from RNA of a cassava plant originally infected with UCBSV-HAM1$_{T1A}$-2xMyc after its digestion with the same restriction enzymes.

The plasmid pLX-UCBSVi-ΔHAM1, which has a full deletion of HAM1 cistron, was built by replacing the above-mentioned BstBI/StuI fragment in pLX-UCBSVi with a compatible end, short, double-stranded DNA fragment created by the annealing of oligonucleotides #3312/#3313.

Plasmids for the expression of recombinant N-terminal MBP-tagged versions of UCBSV HAM1 in *Escherichia coli* were built by the Gateway technology (Invitrogen) using pDONR207 as entry vector and pDEST-TH1 [29] as expression vector. First, pDONR207-HAM1 that carries the coding sequence of the wild-type HAM1 (including a stop codon) was generated by BP recombination between pDONR207 and an amplified PCR fragment generated with primers #3858/#3857 and pLX-UCBSVi as template. Then, mutated versions of HAM1 were produced by overlapping PCR [28], and their corresponding pDONR207-HAM1 derivatives were generated by BP reaction between pDONR207 and PCR fragments. A list of pDONR207-HAM1 derivatives, as well as the name of primers used for the amplification of different inserts, is shown in S3 Table. The correctness of these plasmids was confirmed by digestion with restriction enzymes and Sanger sequencing by Macrogen Europe. Finally, coding sequences corresponding to wild-type and mutated versions of HAM1 were moved from pDONR207 to pDEST-TH1 by LR recombination.

Plasmids for transient expression of viral proteins in *N. benthamiana* leaves were also built by the Gateway technology (Invitrogen), using pENTR1A as entry vector, and either pGWB702Ω (35S promoter, TMV 5'UTR, no tag, NOS terminator) or pGWB718 (35S promoter, 4xMyc tag for N-terminal fusion, NOS terminator) [30] as expression vectors. Briefly, cDNA fragments encoding NIa and $NIb_C$-HAM1-$CP_N$ from UCBSV, CBSV and EuRV were amplified by PCR and directly introduced into pENTR1A previously digested with XmnI/EcoRV (name of primers and templates used for each PCR are indicated in S4 Table). The correctness of pENTR1A derivatives was confirmed by digestion with restriction enzymes and Sanger sequencing by Macrogen Europe. Then, those cDNAs were moved from pENTR1A derivatives to either pGWB702Ω (NIa) or pGWB718 ($NIb_C$-HAM1-$CP_N$) by LR recombination.

### Alignment of primary amino acid sequences and 3D protein modeling

The primary amino acid sequences of the following HAM1 proteins were downloaded from NCBI: human ITPA (NP_258412.1), *E. coli* RdgB (NP_417429.1), yeast HAM1 (NP_012603.1), arabidopsis HAM1-like protein (NP_567410.1), and viral HAM1-like proteins from CBSV (ACS71538.1), UCBSV (ASG92166.1) and EuRV (YP_009310049.1). Protein sequences were aligned with Clustal Omega from EMBL-EBI [31] with default parameters, and results were visualized/colored with Jalview version 2.11.1.4 [32]. The tridimensional structure of UCBSV HAM1 bound to ITP was modeled by homology using the SWISS-MODEL server [33].

### Purification of recombinant HAM1 variants produced in *E. coli* and measurement of their ITPase activity

The expression of recombinant MBP-tagged HAM1 variants was carried out in BL21 (pLysS) strain of *E. coli*. 100-ml cultures were grown at 37°C to $OD_{600}$ of 0.6–0.8 with constant shaking at 250 rpm. Recombinant protein expression was induced with isopropyl β-D-thiogalactopyranoside (IPTG) to a final concentration of 1 mM during 4 hours at 30°C. Cells were harvested by centrifugation and stored at −80°C until used. Frozen pellets were thawed and lysed by sonication in 5 ml of buffer A (20 mM Tris-HCl pH = 8 and 150 mM NaCl) in the presence of cOmplete™ EDTA-free protease inhibitor (Roche). Cell debris was separated by centrifugation, and each supernatant was incubated with 400 μl of previously equilibrated amylose resin (New England Biolabs) in Poly-Prep chromatography columns (Bio-Rad) under rotation for 2 hours at 4°C. Columns were placed vertically and resin was settled by gravity for a few minutes. After supernatants flowed out, columns were washed with 10 ml of buffer A. Finally, proteins were eluted with buffer A + 10 mM maltose. ITPase activity of purified MBP-HAM1 variants was estimated with the PRECICE ITPase assay kit by following the manufacturer instructions (Nobocib). HAM1 activity was calculated as indicated by the manufacturer (indicated also in S2A Fig).

### Virus inoculation

Inoculation of UCBSV full-length clones (wild type and derivatives) was carried out by biolistic with the Helios Gene Gun System (Bio-Rad) by following a previously described protocol [34]. Helium pressures of 7 and 13 bar were used to inoculate *N. benthamiana* and cassava, respectively. Serial passages were done by manual inoculation of plants with sap extracts from infected plants as viral source. To do that, infected leaves were ground in a buffer containing 150 mM NaCl, 2.5 mM DTT and 50 mM Tris-HCl pH 7.5 (2ml/mg) with an ice-cold mortar

and pestle, and the sap was finger-rubbed onto two leaves of plants that had previously been dusted with Carborundum.

## Fluorescence imaging

GFP fluorescence was observed with an epifluorescence stereomicroscope using excitation and barrier filters at 470/40 nm and 525/50 nm, respectively, and photographed with an Olympus DP70 digital camera.

## Transient expression by agro-infiltration

Two young leaves of 1-month-old *N. benthamiana* plants were infiltrated with *Agrobacterium tumefaciens* strain C58C1 carrying the indicated plasmids, as previously described [9]. To boost protein expression, the potent silencing suppressor P14 from photos latent virus [35] was co-expressed along with the proteins of interest.

## Immunodetection of proteins by western blot

The preparation of protein samples under denaturing conditions, the separation on SDS-PAGE and the electroblotting to nitrocellulose membranes was previously described [36]. UCBSV was detected using anti-CP (Ref. AS-1153, DSMZ) as primary antibody and horseradish peroxidase (HRP)-conjugated goat anti-rabbit IgG (Ref. 111-035-003, Jackson ImmunoResearch) as the secondary reagent. GFP and Myc-tag were detected using anti-GFP (Ref. 11814460001, Roche) and anti-Myc (either Ref. M20002, AbMART; or Ref. 05–724, Millipore) as primary antibodies, respectively, and HRP-conjugated sheep anti-mouse IgG (Ref. NA931, Amersham) as the secondary reagent. Immunostained proteins were visualized by enhanced chemiluminescence detection with Clarity ECL Western blotting substrate (Bio-Rad) in a ChemiDoc apparatus (Bio-Rad). Ponceau red was used to verify equivalent loading of total proteins in each sample.

## Reverse transcription followed by PCR

Firstly, total RNA was isolated from *N. benthamiana* and cassava leaves by using the FavorPrep Plant Total RNA Purification Mini Kit (Ref. FAPRK 001, Favorgen Biotech) and Spectrum Plant Total RNA Kit (Ref. STRN50, Sigma), respectively. The RNA integrity was verified by electrophoresis in agarose gel. Secondly, cDNA was synthesized from 500 ng of total RNA with retrotranscriptase from Moloney murine leukemia virus (Ref. M0253, New England BioLabs) and random hexanucleotides as primers by following the manufacturer's instructions. Then, the cDNAs were used as template to amplify the region that encodes $NIb_C$-HAM1-$CP_N$ with primers #3160/#3130 or the one that encodes a short fragment of CP with primers #3547/#3130. In the particular case of samples from cassava, which are prone to be contaminated with RT-PCR inhibitors such as polyphenols, RNA quality was checked by RT-PCR amplification of the UBQ10 housekeeping gene [37] in order to validate the lack of amplification of UCBSV-derived fragments in samples from non-infected plants. Finally, when required, PCR products were Sanger sequenced by Macrogen Europe.

## Measurement of NTPs in plant leaves

Free NTPs were extracted from young leaves of *N. benthamiana* and cassava by using a previously described method [38]. Extracts were immediately injected into a Vanquish UHPLC system equipped with a Q Exactive Focus Orbitrap spectrometry detector (Thermo Fisher Scientific). NTPs were separated by means of a Primesep SB column (3 μm, 4.6 x 150 mm)

(SIELC Technologies) with a mobile phase formed by a mixture of (A) acetonitrile and water (5/95 v/v) with 30 mM of ammonium acetate (pH 4.5) and (B) acetonitrile and water (10/90 v/v) with 200 mM of ammonium acetate (pH 4.5) flowing at 1.0 ml/min with a gradient from 50-to-100% of A in 15 minutes. Injection was set to 5 μl and the column temperature to 25˚C. Electrospray ionization was done at 4000V, setting the capillary temperature to 400˚C. Desolvation was carried out with nitrogen with sheath gas and auxiliary gas flow rates of 70 and 20 (500˚C), respectively. NTPs were detected in MS/MS experiments (scan range from 50 to 550) based on the transition from the molecular protonated cation ([M+H]+) to the breakdown product consisting of the corresponding protonated nucleobase ([Nb+H]+) at collision energy of 25 eV.

## Results

### UCBSV does not require HAM1 to infect *N. benthamiana*

With the aim of tracking the UCBSV infection *in planta*, an UCBSV full-length cDNA clone was manipulated to introduce the GFP coding sequence between NIb and CP cistrons (Fig 1A). The infection efficiency of UCBSV-eGFP was compared with that of the wild-type UCBSV in the model plant *N. benthamiana*. Plants inoculated with the wild-type virus (n = 3) started to display clear symptoms of viral infection in upper non-inoculated leaves at 10 dpi, whereas those inoculated with UCBSV-eGFP (n = 3) had a delay in symptom appearance of 2-to-3 days. At 15 dpi all inoculated plants showed equivalent symptoms in apical leaves, including strong leaf curling and vein clearing (Fig 1B); however, in line with the delay in symptom appearance, the height of plants inoculated with UCBSV-eGFP was in the middle of the untreated plants (tallest) and those infected with the wild-type UCBSV (shortest) (Fig 1B). As expected, upper non-inoculated leaves of plants infected with UCBSV-eGFP displayed GFP-derived fluorescence when observed under UV light (Fig 1B). In accordance with the other infection parameters, viral load in upper non-inoculated leaves, inferred from UCBSV CP immunodetection, was slightly higher in plants infected with the wild-type virus (Fig 1C). The immunodetection analysis also showed that GFP produced by UCBSV-eGFP was properly released from the viral polyprotein during the infection (Fig 1C).

After a plant-to-plant passage, unlike in plants initially inoculated with cDNA clones, we observed no differences among plants infected with wild-type and GFP-tagged viruses regarding the time of appearance and intensity of systemic symptoms. When upper non-inoculated leaves of plants infected with UCBSV-eGFP were observed under UV light at 20 dpi, curiously, fluorescence was not detected, suggesting that the GFP cistron had been deleted from the viral genome. Indeed, a deeper analysis of viral populations from these plants confirmed this assumption, as DNA products amplified by RT-PCR with primers flanking the HAM1-GFP coding region were much smaller than those produced from plants originally infected by shooting (Fig 1D). Remarkably, direct Sanger sequencing of these products showed that not only GFP-, but also HAM1-coding sequences, had been either totally or partially deleted from UCBSV-eGFP after the first passage (Fig 1D). This result support the idea that HAM1 is not required for the virus to infect *N. benthamiana*.

To confirm that HAM1 is unnecessary for UCBSV to infect the experimental host *N. benthamiana*, and rule out the possibility that a small fraction of the wild-type virus was complementing the deletion mutant, we built an infectious cDNA clone that carries a complete deletion of HAM1 cistron (Fig 2A). *N. benthamiana* plants inoculated with plasmids expressing either UCBSV or UCBSV-ΔHAM1 (n = 3 per construct) started to display clear infection symptoms in upper non-inoculated leaves at the same time, and these symptoms were similar in intensity and type (Fig 2B). Estimation of viral load in these plants was carried out in samples from systemically infected leaves by RT-qPCR to detect small differences, if any. As

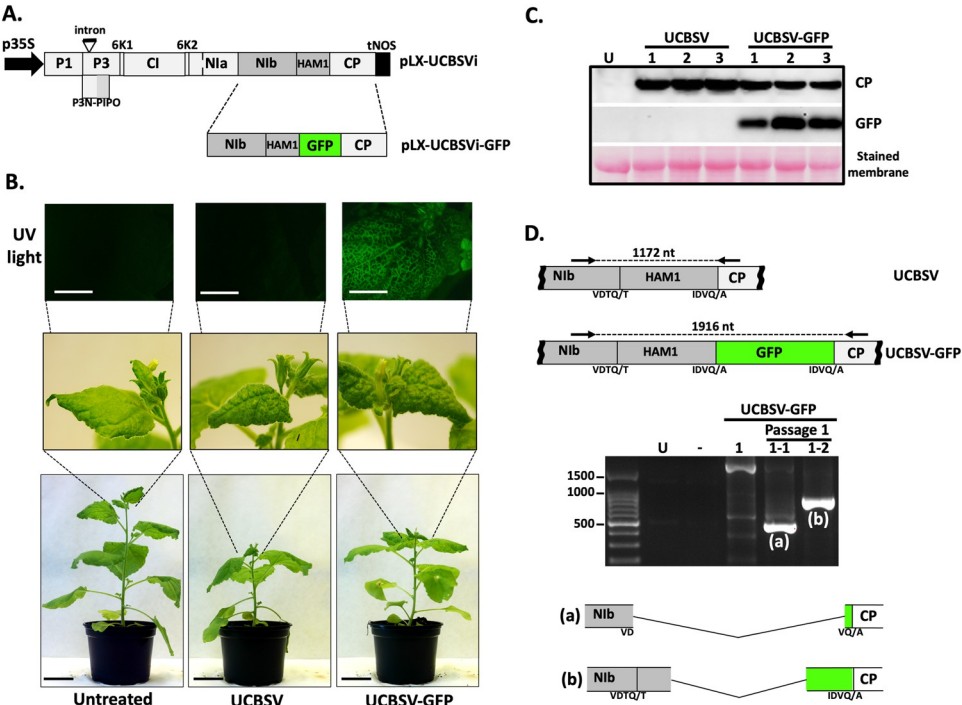

**Fig 1. GFP-tagged UCBSV loses HAM1 and GFP coding sequences after one passage in *N. benthamiana*.** (A) Schematic representation of viral constructs based on the pLX-UCBSV [25] used in this experiment. Boxes represent mature viral factors as they are encoded in the viral genome. The presence of an intron in the P3 coding sequence is also indicated. p35S: 35S promoter from cauliflower mosaic virus; tNOS: terminator from the NOS gene of *Agrobacterium tumefaciens*. (B) Representative pictures taken at 15 days post-inoculation of infected and non-treated *N. benthamiana* plants under UV radiation and visible light (white bar = 1 cm; black bar = 4 cm). (C) Detection of GFP and UCBSV CP by immunoblot analysis in protein samples from upper non-inoculated leaves of *N. benthamiana* plants infected with the indicated viruses. Blot stained with *Ponceau* red showing the large subunit of the ribulose-1,5-bisphosphate carboxylase-oxygenase is included as a loading control (D) Agarose gel electrophoresis analysis of a viral genomic fragment amplified by RT-PCR from plants infected with UCBSV-GFP after one passage. The upper part shows a schematic representation of the amplified fragment. Black arrows represent primers used for amplification. Sizes of expected PCR products are indicated. Amino acids around the NIaPro cleavage sites are depicted at the bottom.

observed in Fig 2C, accumulation of viral RNA did not show significant differences between both viruses. Together, experiments shown in Figs 1 and 2 demonstrate that HAM1 is not required to produce an UCBSV wild-type-like infection in *N. benthamiana*.

## UCBSV requires HAM1 pyrophosphatase activity to infect its natural host

Based on the above results, we hypothesized that the presence of HAM1 in UCBSV is a specific requirement for the virus to infect its natural host. To test this guess, we inoculated cassava and *N. benthamiana* plants in parallel with UCBSV and UCBSV-ΔHAM1 (n = 3 per virus and plant species). As in the previous experiment, *N. benthamiana* plants displayed clear symptoms of UCBSV infection at 9-to-10 dpi in upper non-inoculated leaves independently of the presence/absence of HAM1 cistron in the viral genome. Cassavas, in turn, developed typical UCBSV symptoms (yellow mottling along the major veins in leaves and dark brown streaks in stems) by 45-to-60 dpi in plants inoculated with the wild-type virus (Fig 2D). In contrast, plants inoculated with UCBSV-ΔHAM1, as those untreated, had normal leaf coloring and lacked streaks in stems (Fig 2D), even after 180 dpi. The presence of UCBSV in these plants was tested by RT-PCR in samples collected at 60 dpi from upper non-inoculated leaves. The

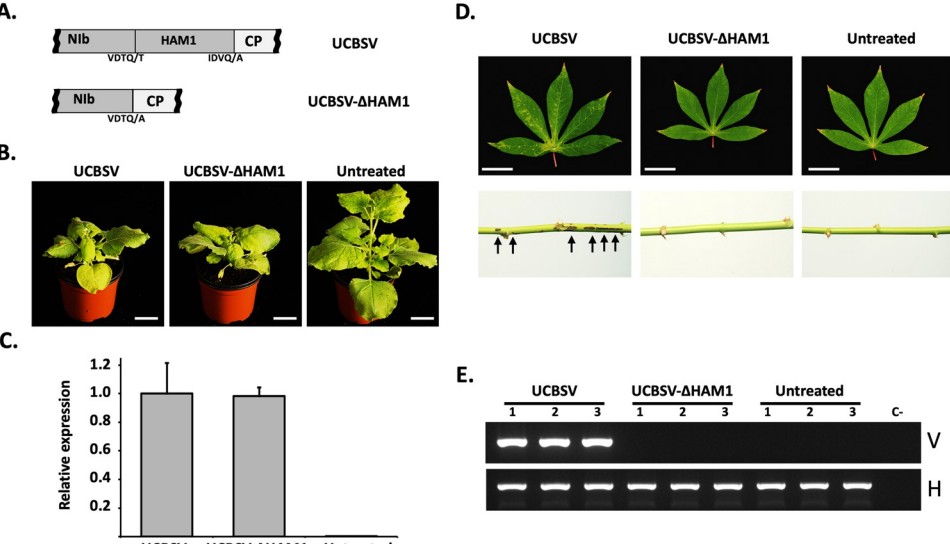

**Fig 2. Virus-derived HAM1 is required for the successful infection of UCBSV in cassava plants, but not in *N. benthamiana*.** (A) Schematic representation of the NIb-to-CP genomic segment of viruses used in these experiments. Amino acids around the NIaPro cleavage sites are depicted. (B) Representative pictures of infected and non-treated *N. benthamiana plants* taken at 12 days post-inoculation. White bar = 4 cm. (C) RT-qPCR measuring the accumulation of viral RNA in upper non-inoculated leaves of *N. benthamiana plants* infected with the indicated viruses. Each bar represents the average of three plants (error bar = 1 standard deviation). For normalization, the average of wild type UCBSV is equal to 1. (D) Representative pictures of upper non-inoculated leaves and stems, at 60 days post-inoculation, of cassava plants inoculated with the indicated viruses. White bar = 4 cm. Black arrows indicates the presence of brown streaks in the stem of an infected plant. (E) Analysis by agarose gel electrophoresis of a fragment of the UCBSV genome (V) and of a plant housekeeping gene (H) amplified by RT-PCR. RNA samples from upper non-inoculated leaves of 3 independent cassava plants inoculated with the indicated viruses were used as template.

result confirmed our visual observation: only plants inoculated with wild-type UCBSV accumulated viral RNA in upper non-inoculated tissues (Fig 2E).

Our results suggested that UCBSV requires a pyrophosphatase activity to infect cassava. In order to test this hypothesis, and to rule out the possibility that the lack of infectivity of UCBSV-ΔHAM1 in cassava was rather due to an undesired side effect caused by the deletion of the whole HAM1 cistron from the viral genome, we aimed to introduce just a single point mutation in HAM1 to specifically disrupt its pyrophosphatase activity. Based on previous reports on the crystal structure of the human HAM1 (named ITPA) bound to ITP [39], we modeled with high confidence (QMEAN = -0.66) the tridimensional conformation of a UCBSV HAM1 dimer bound to this non-canonical nucleotide (Fig 3A). The K amino acid at position 38, which is located in the second α-helix, is among the fully conserved amino acids in HAM1-like proteins from potyvirids and from organisms as diverse as *E. coli*, baker's yeast, *Arabidopsis* and human (S1 Fig). In ITPA, this particular K (K19) is proposed to be part of the protein catalytic centre, as its side chain directly interacts with the triphosphate group of ITP [39] (Fig 3A). Moreover, in line with such relevance, a mutation of this K (K13) in RdgB, the HAM1-like protein from *E. coli*, abolishes its capacity to hydrolyze ITP *in vitro* [40]. To experimentally examine the relevance of K38 for the ITPase activity of UCBSV HAM1, we produced both wild-type and K38A mutant versions of MBP-tagged HAM1 in *E. coli* and measured their ITPase activity in a coupled reaction (S2A Fig) by using equivalent amounts of purified proteins (S2B Fig). As anticipated from previous published results with a 6xHis-tagged version of UCBSV HAM1 [24], MBP-HAM1 hydrolyzed ITP (S2C Fig). More importantly, as predicted above, the change of K38 by alanine completely abolished this activity (S2

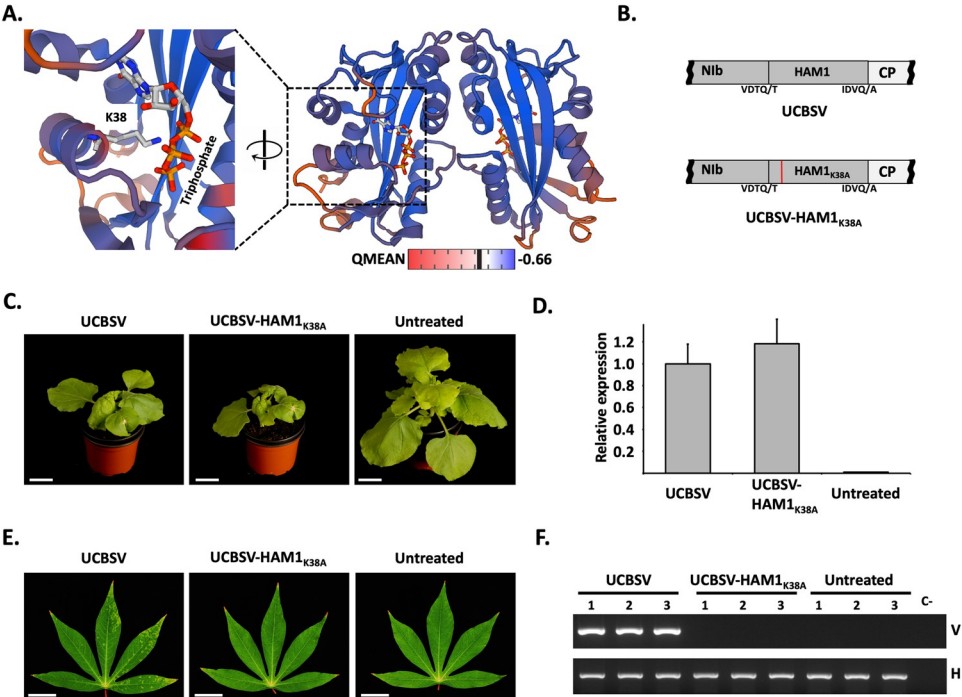

**Fig 3. The pyrophosphatase activity of UCBSV HAM1 is required for the successful infection of cassava plants.**
(A) Model of the ITP-bound UCBSV HAM1 tridimensional structure. Interaction between K38 and ITP is highlighted at the left. (B) Schematic representation of the NIb-to-CP genomic segment of viruses used in these experiments. Amino acids around the NIaPro cleavage sites are depicted. The presence of the K38A mutation is indicated with a red line. (C) Representative pictures of infected and non-treated *N. benthamiana plants* taken at 11 days post-inoculation. White bar = 4 cm. (D) RT-qPCR measuring the accumulation of viral RNA in upper non-inoculated leaves of *N. benthamiana plants* infected with the indicated viruses. Each bar represents the average of three plants (error bar = 1 standard deviation). For normalization, the average of wild type UCBSV is equal to 1. (E) Representative pictures of upper non-inoculated leaves, taken at 60 days post-inoculation, of cassava plants inoculated with the indicated viruses. White bar = 4 cm. (F) Analysis by agarose gel electrophoresis of a fragment of the UCBSV genome (V) and of a plant housekeeping gene (H) amplified by RT-PCR. RNA samples from upper non-inoculated leaves of 3 independent cassava plants inoculated with the indicated viruses were used as template.

Fig). Based on these data, we build an UCBSV cDNA clone that carries the mutation K38A in HAM1 (Fig 3B). The wild-type and mutant versions of UCBSV were inoculated in *N. benthamiana* and cassava in parallel (n = 3 per virus and plant species). As expected, there were no differences in *N. benthamiana* plants inoculated with each of these viruses in infectivity, time of appearance and intensity of symptoms in upper non-inoculated leaves (Fig 3C), as well as in viral accumulation measured by RT-qPCR in samples from these tissues (Fig 3D). Conversely, only the three cassava plants inoculated with the wild-type virus displayed symptoms of viral infection in upper non-inoculated leaves at 60 dpi (Fig 3E). Further analysis by RT-PCR confirmed that the wild-type UCBSV, but not the mutant variant that carries the K38A mutation in HAM1, was able to infect cassavas (Fig 3F). Together, these results indicate that an active pyrophosphatase contributes to UCBSV infection, and the requirement of this activity depends on the particular host.

## UCBSV that carries a partial loss-of-function mutation in HAM1 displays lower fitness than the wild-type virus and evolves to gain ITPase activity

To further study the dependency of UCBSV on HAM1 pyrophosphatase activity to infect cassava plants, we aimed to introduce a mutation that diminishes this activity to certain extend

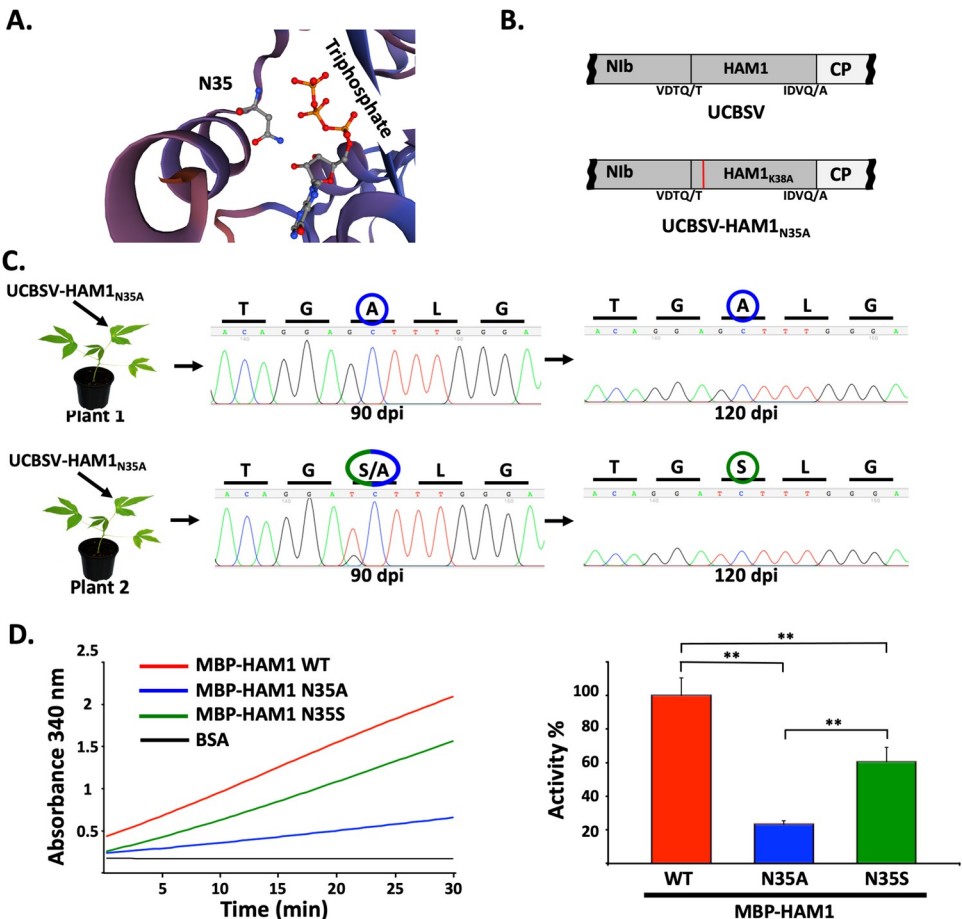

**Fig 4. Evolution of a HAM1 partial loss-of-function UCBSV mutant in cassava.** (A) Zoom on the model of UCBSV HAM1 tridimensional structure to highlight the location of the asparagine at position 35 inside the catalytic pocket. (B) Schematic representation of the NIb-to-CP genomic segment of viruses used in these experiments. Amino acids around the NIaPro cleavage sites are depicted. The presence of the N35A mutation is indicated with a red line. (C) Chromatograms of Sanger sequencing results of the DNA fragment of interest amplified by RT-PCR. RNA samples deriving from upper non-inoculated leaves of the indicated cassava plants inoculated with UCBSV-HAM1N$_{35A}$ were used as template. Leaves for RNA preparation were harvested at both 90 and 120 days post-infection (dpi). Colored circles surround both the original and the naturally introduced mutation. (D) Left panel: kinetics of ITP hydrolysis, measured by the formation of NADH$_2$ as indicated in S2A Fig, when 10 nanograms of the indicated proteins were used in the reaction mix. Right panel: bar graph that represents averages of ITPase specific activity when different amount of the indicated protein were used (n = 3; 10, 20 and 40 nanograms, error bar = 1 standard deviation). For normalization, the average activity of wild-type HAM1 is equal to 100%. Statistical differences were tested with the post-hoc Tukey HDS test (** p < 0.01).

without complete abolishment. We predicted that this mutant should infect cassava with lower fitness than the wild-type virus. Based on previously published protein structures [39,40], we decided to mutate the conserved asparagine at position 35 (N10 in RdgB and N16 in ITPA, S1 Fig), since it seems to be part of the catalytic center (Fig 4A), and because changing this amino acid by alanine in RdgB reduces its activity to 30% of that of the wild-type protein [40]. Therefore, we inoculated *N. benthamiana* and cassava plants (n = 3 per virus and plant species) with the wild-type UCBSV infectious clone (positive control) or with the plasmid that carries the HAM1 N35A mutation (Fig 4B). As expected, no differences were observed in *N. benthamiana* plants inoculated with these two viruses in term of infectivity and viral symptoms (S3A Fig). In contrast, only those cassava plants inoculated with the wild-type virus developed symptoms of viral infection in upper non-inoculated leaves at 40 dpi (S3B Fig). As late as 90 dpi, some

symptoms of viral infection appeared in stems of the three cassava plants inoculated with UCBSV-HAM1$_{N35A}$, although these symptoms were fainter than those in plants infected with the wild-type virus (S3C Fig). RT-PCR confirmed that upper non-inoculated leaves from these plants were successfully infected, whilst Sanger sequencing analysis of these products revealed that the introduced mutations were maintained (Figs 4C and S3D, 90 dpi), except in plant number two, as the alanine at position 35 of HAM1 had changed to serine (GCT -> TCT) in part of the viral population (Fig 4C, 90 dpi). Remarkably, at 120 dpi, UCBSV-HAM1$_{N35A}$ was overcome by UCBSV-HAM1$_{N35S}$ (serine at position 35) in plant two (Fig 4C, 120 dpi), as well as in plant number three (S3D Fig, 120 dpi).

Based on these results, and having in mind the above proposed relevance of HAM1 pyro-phosphatase activity for infection in cassava, we hypothesized that UCBSV-HAM1$_{N35A}$ evolved to UCBSV-HAM1$_{N35S}$ because HAM1$_{N35S}$ has higher pyrophosphatase activity than HAM1$_{N35A}$. To test this idea, we produced both N35A and N35S mutant versions of MBP-tagged HAM1 in *E. coli*, and equal amounts of proteins were used to measure their ITPase activity along with the wild-type MBP-tagged protein (S3E Fig), used here as reference (100% activity). As shown in Fig 4D, all three proteins hydrolyzed ITP, but they did it with distinct efficiency. Indeed, measurement of the rate of ITP conversion to NAD showed that N35A variant had around 20% activity of that of the wild-type protein, whereas N35S had around 60%. Therefore, this experiment shows that a virus with a partial loss-of-function mutation in HAM1 evolves to gain ITPase activity. All in all, results presented in Figs 3 and 4 highlight the importance of the non-canonical nucleotide hydrolysis mediated by HAM1 for UCBSV to infect its natural host.

## Differential accumulation of NTPs in *M. esculenta* versus *N. benthamiana*

Our observation that pyrophosphatase activity is only required for UCBSV infection in cassava prompted us to investigate the accumulation of canonical and non-canonical nucleotides in *M. esculenta* and *N. benthamiana* plants. To do that, NTPs were extracted from equivalent amount of tissue powder from both UCBSV hosts (n = 12 per plant species) and the relative concentrations of ATP, CTP, GTP, UTP, ITP and XTP were estimated by high performance liquid chromatography coupled with tandem mass spectrometry. Whereas the concentration (measured as the area under the curve) corresponding to CTP was equivalent in both plants, showing no significant difference, that of XTP, ITP, GTP and UTP and ATP were significantly higher in cassava relative to *N. benthamiana* (Fig 5). Differences in nucleotide accumulation were particularly relevant in the case of the non-canonical nucleotides XTP (4.5 folds) and ITP (3.6 folds) (Fig 5). Importantly, an independent repetition of this experiment showed equivalent differences when comparing the population of NTPs in leaves of these two plant species. Therefore, we can conclude that *M. esculenta*, the natural host of UCBSV, accumulates much higher levels of XTP and ITP in leaves than the *N. benthamiana* counterpart.

## Suboptimal cleavage at NIb/HAM1 junction during UCBSV infection

When the presence of HAM1 cistron in the genome of UCBSV (named CBSV at that time) was reported for the first time, authors proposed that NIb and HAM1 might accumulate as two independent mature factors in infected cells due to the presence of a putative target for the viral-derived protease NIaPro [11]. A canonical NIaPro cleavage site is formed by 9 moder-ately conserved amino acids, and cleavage occurs between residues 6 and 7 (P1 and P1', Fig 6A) [41]. Amino acid primary sequence analysis shows that positions P4, P1 and P1' have high degree of conservation. For P1', for instance, either A, S or G residues were observed in 84% of the cases (n = 1154, from 165 viral genomes) [42]. Curiously, a T residue occupies this position

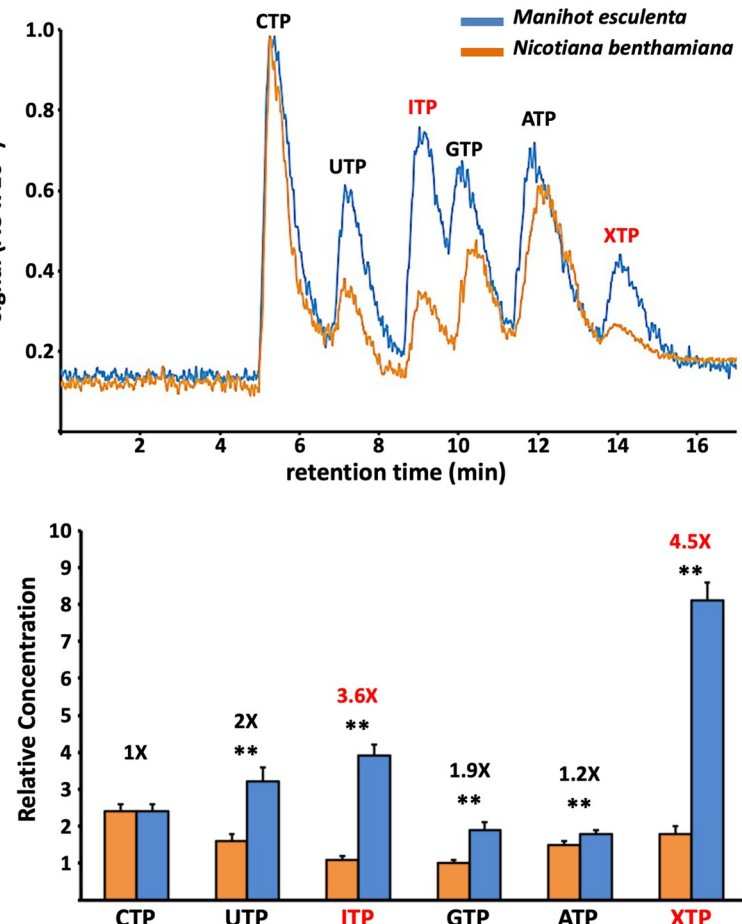

**Fig 5. High accumulation of non-canonical nucleotides in cassava.** Upper panel: base peak chromatogram in arbitrary units (AU) for representative samples of total NTPs from *Manihot esculenta* (blue) and *Nicotiana benthamiana* (orange) analysed accordingly to the method described in the text. Lower panel: Bar graph that represents averaged concentrations (n = 12; for normalization, GTP in *N. benthamiana* = 1). The original data set showing relative concentrations is presented in S5 Table. Statistical differences were tested with the post-hoc Tukey HDS test (** $p < 0.01$). Ratios (*M. esculenta/N. benthamiana*) for each NTP are also indicated. Non-canonical nucleotides are highlighted in red.

in the cleavage site located at the NIb/HAM1 junction of UCBSV (Fig 6A), which is not a common amino acid at P1' with a representation of 1.5% [42]. In fact, a seminal study about the NIaPro-mediated cleavage at the NIb/CP junction of tobacco each virus, another *Potyviridae*, showed that S x T mutation at P1' strongly reduced cleavage efficiency in an *in vitro* system [see Fig 4D in [43]].

The above-mentioned antecedents prompted us to investigate whether the proposed cleavage site located between NIb and HAM1 is efficiently processed during UCBSV infection. To do that we built an infectious cDNA clone in which HAM1 was tagged with two copies of the Myc epitope (UCBSV-HAM1-2xMyc, Fig 6B) for the easy detection of HAM1 in extracts of infected tissues. This clone, and the clone that expresses the wild-type UCBSV as control, were inoculated in *N. benthamiana* plants (n = 3 per virus). No differences among inoculated plants were observed in term of viral symptoms (Fig 6C) and accumulation as estimated by western blot against UCBSV CP (Fig 6D). This result indicated that the tag does not have a noticeable negative impact on viral fitness in *N. benthamiana*. Immunodetection with Myc antibody revealed the presence of two defined protein species in samples infected with UCBSV-2xMYC.

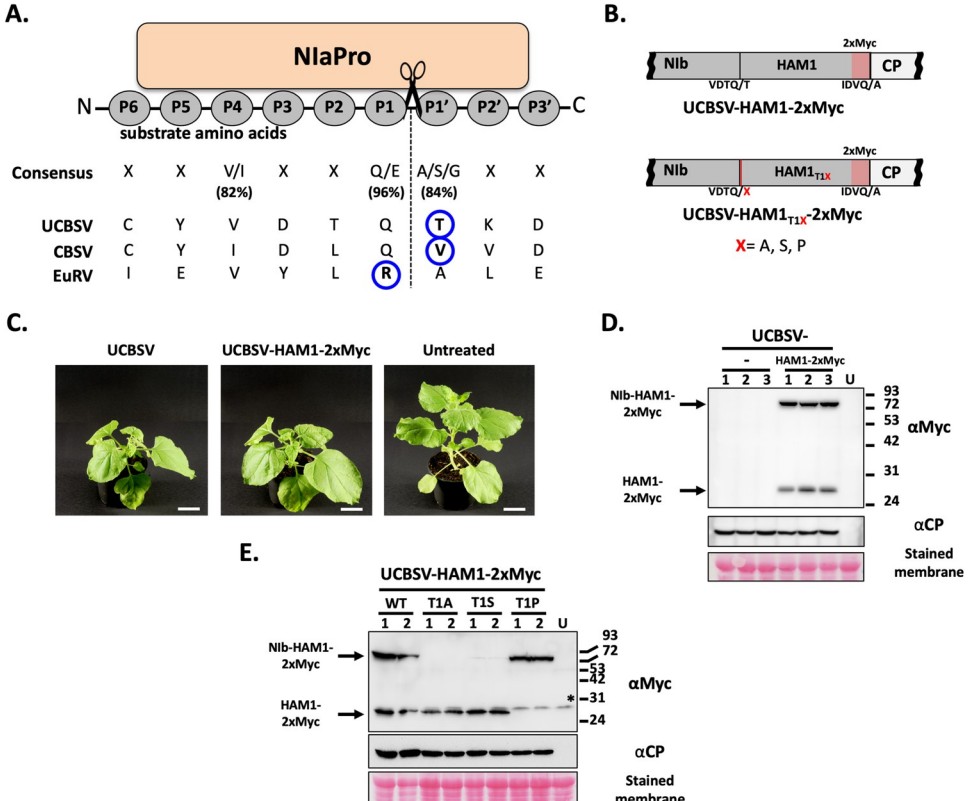

**Fig 6. Suboptimal separation of NIb-HAM1 during UCBSV infection.** (A) Schematic representation of a NIaPro cleavage site. Substrate residues at both sides of the scissile bond are labeled by following a previously proposed nomenclature [52]. The consensus sequence of NIaPro substrates, as well as those residues present at the NIb-HAM1 junction in UCBSV, CBSV and EuRV, are indicated. Percentages at which conserved amino acids were found at the indicated positions are shown (n = 1154) [42]. The non-conserved residue at the NIb-HAM1 cleavage site of each virus is surrounded by a blue circle. (B) Schematic representation of the NIb-to-CP genomic segment of viruses used in these experiments. Amino acids around the NIaPro cleavage sites are depicted. (C) Representative pictures of infected and non-treated *N. benthamiana plants* taken at 13 days post-inoculation. White bar = 4 cm. (D and E) Detection of Myc-tagged HAM1 and CP by immunoblot analysis in samples from upper non-inoculated leaves of *N. benthamiana* plants infected with the indicated viruses. The positions of prestained molecular mass markers (in kilodaltons) run in the same gels are indicated to the right. The black asterisk indicates the presence of a cross-reacting band in all the samples, including the untreated control. Blots stained with *Ponceau* red showing the large subunit of the ribulose-1,5-bisphosphate carboxylase-oxygenase are included as a loading control.

The one with less electrophoretic mobility had the expected size for the Myc-tagged NIb-HAM1 fusion product (86.3 kDa), whereas the smaller species had the expected size for the sole Myc-tagged HAM1 (28.2 kDa) (Fig 6D). The ratio between larger and smaller species was estimated in 1.5 based on the densitometric analysis of chemiluminescence signals.

Our results, along with previous antecedents (see above), suggested that T at position P1' causes an inefficient NIaPro-mediated processing at the cleavage site located in the NIb/HAM1 junction. To test this idea, we introduced mutations in the UCBSV cDNA clone to express two types of P1' mutants: (i) T1A and T1S, as A and S are among the most frequent residues at this position, and (ii) TxP, as P is not present at the P1' position in any NIaPro cleavage site [42]. When mutated and wild-type versions of UCBSV-2xMYC were inoculated in *N. benthamiana* (n = 2 per virus), all of them produced indistinguishable infections, with comparable symptoms (S4 Fig) and virus accumulation in upper non-inoculated leaves as observed by immunodetection of UCBSV CP (Fig 6E). As anticipated from conservation of amino acids present at the P1' position, the T1A and T1S mutants accumulated only the

protein species that corresponds to free HAM1, while T1P mutant only produced the NIb-HAM1 complex. Altogether, these results indicate that the NIaPro-mediated separation of NIb and HAM1 is inherently inefficient in UCBSV, which is due to the presence of a T residue at the P1' position of the cleavage site.

## Relevance of the inefficient cleavage at NIb/HAM1 junction in cassava

To estimate the relevance of the poor separation of NIb from HAM1 in the UCBSV natural host, we inoculated cassava plants with the Myc-tagged wild-type virus as well as the T1A and T1P variants (n = 3 per virus). Clear symptoms of infection appeared at 60 dpi in the upper leaves of all inoculated plants independently of the infecting virus (Fig 7A). At that time, RT-PCR confirmed that upper non-inoculated leaves from all inoculated plants were successfully infected with the Myc-tagged viruses (Fig 7B). Moreover, Sanger sequencing analysis of these RT-PCR products indicated that the introduced mutations were maintained after two months (Fig 7C). At 120 dpi, we divided plants infected with each virus in two groups, such as one plant was kept growing (plant 1), whereas the remaining two plants were propagated through stem cuttings (plant 2 and plant 3). At 180 dpi, samples were taken from the upper leaves of all plants and the identity of infecting viruses was determined. Remarkably, whereas the NIb-HAM1 junction from both wild-type and the T1P variants remained unchanged in all the analyzed plants, that of the T1A variant evolved to introduce mutations (Figs 7C and S5). In the case of plant 1, a second mutation appeared at the P3' position, so that the original D amino acid was replaced by G (D3G) to give rise to a T1A/D3G double mutant (Fig 7C). Cuttings made from plant 2 also accumulated a viral variant carrying the second mutation G3D, reinforcing the idea that the T1A single mutant evolves to T1A/D3G when adapting to cassava (S5 Fig). Finally, cuttings made from plant 3 showed accumulation of a variant that encodes the wild-type cleavage site, so that the T1A mutation reverted to T (S5 Fig).

The observed reversion in cuttings from plant 3 strongly suggested that the virus requires the inefficient processing of the cleavage site located at the NIb-HAM1 junction for a successful infection. If that were the case, then one would expect that the T1A/D3G double mutant mimics this phenotype. To test this idea, we built the double mutant T1A/D3G by directed mutagenesis of the UCBSV-HAM1-2xMyc clone, and this plasmid was used to inoculate *N. benthamiana* plants (n = 4) for easy detection of processing products by western blot. Both UCBSV-HAM1-2xMyc and UCBSV-HAM1$_{T1A}$-2xMyc variants were used as control (n = 2 per variant). As expected, the T1A/D3G double mutant behaved as controls in term of infection timing and visible symptoms, as well as viral accumulation in upper non-inoculated leaves as estimated by immunodetection of UCBSV CP (Fig 7D). Detection of Myc-tagged proteins in samples from systemically infected tissue showed that the T1A/D3G double mutant, as in the case of the wild-type virus, and unlike the T1A variant, accumulated two different protein species: NIb-HAM1 and free HAM1 (Fig 7D). Therefore, our results indicate that, at least for a relevant fraction of the total NIb and HAM1 produced during UCBSV infection, (i) these two proteins stay covalently bound, and (ii) the NIb-HAM1 partnership is indeed relevant when UCBSV infects its natural host. In addition, our viral evolution experiment highlights the importance of the usually underestimated amino acids located at P3' position of NIaPro cleavage sites for the actual NIaPro processing.

## The expression of a joint NIb-HAM1 product is a common future of potyvirds encoding HAM1

CBSV and EuRV are also potyvirds encoding HAM1 in their genomes, and this cistron is located, as in the case of UCBSV, just downstream of NIb. Importantly, the previously

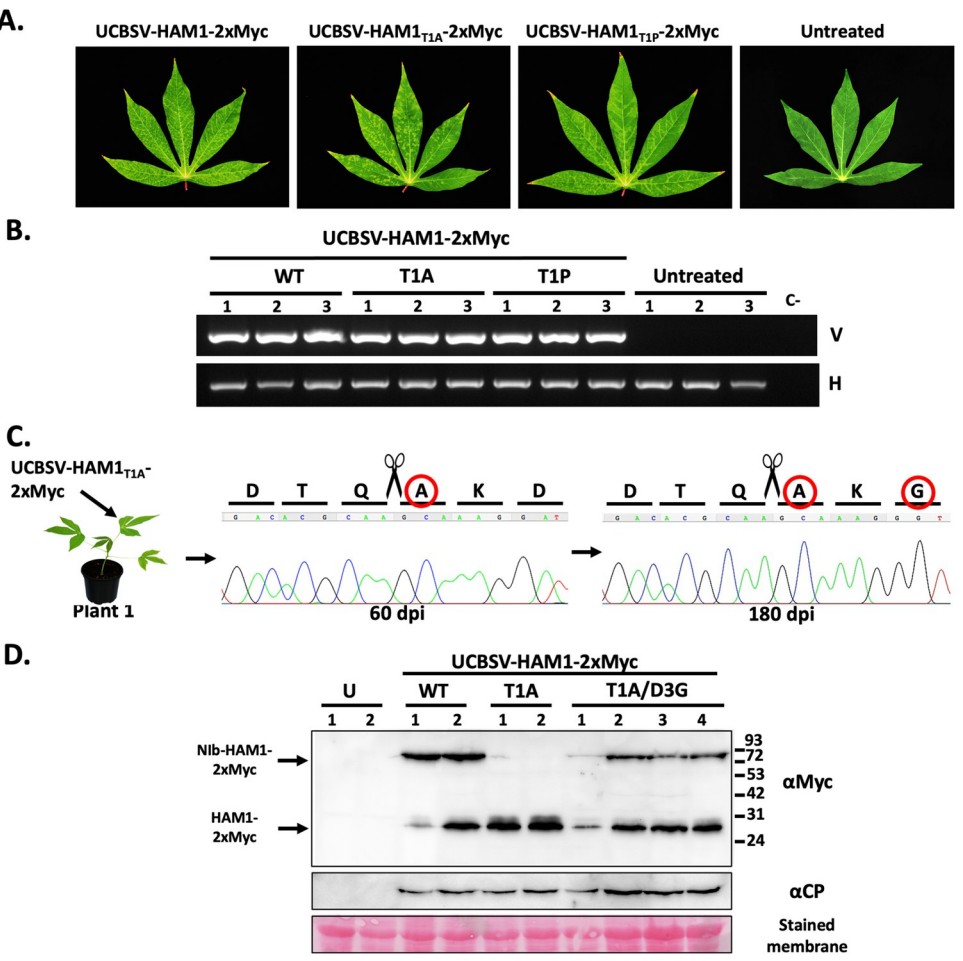

**Fig 7. UCBSV HAM1$_{T1A}$ mutant, which undergoes an optimal cleavage at the NIb-HAM1 junction, evolves to display partial split.** (A) Representative pictures of upper non-inoculated leaves, taken at 60 days post-inoculation, of cassava plants inoculated with the indicated 2xMyc-tagged versions of UCBSV. White bar = 4 cm. (B) Analysis by agarose gel electrophoresis of a fragment of the UCBSV genome (V) and a plant housekeeping gene (H) amplified by RT-PCR. RNA samples from upper non-inoculated leaves of 3 independent cassava plants inoculated with the indicated viruses and collected at 60 dpi, were used as template. (C) Chromatograms of Sanger sequencing results of the UCBSV genomic fragment of interest amplified by RT-PCR. RNA samples from upper non-inoculated leaves of a cassava plant inoculated with the UCBSV-HAM1$_{T1A}$-2xMyc mutant were used as template. Leaves for RNA preparation were harvested at 60 and 180 dpi. Residues derived from the original mutation and from the spontaneous second mutation are surrounded by a red circle. (D) Detection of Myc-tagged HAM1 and UCBSV CP by immunoblot analysis in samples from upper non-inoculated leaves of *N. benthamiana* plants infected with the indicated viruses. The positions of prestained molecular mass markers (in kilodaltons) run in the same gel is indicated to the right. Blot stained with Ponceau red showing the large subunit of the ribulose-1,5-bisphosphate carboxylase-oxygenase is included at the bottom as a loading control.

proposed NIaPro cleavage site at the NIb/HAM1 junction in both viruses [11] does not fit the conventional conservation rules. CBSV has a V at P1', which is not a common residue at this position with a representation of 1.4% [42] (Fig 6A). In the case of EuRV, P1 is occupied by R, which is a strongly underrepresented amino acid at this position with a representation of 0.3% [42] (Fig 6A). Therefore, we hypothesized that HAM1 also remains bound to CBSV and EuRV NIbs. To test this idea, and due to the lack of infectious cDNA clones for these two viruses, we transiently expressed 4xMyc c-terminal tagged versions of NIb-HAM1-CP either with or without the presence of their cognate NIa (VPg-NIaPro) proteinases (Fig 8A). We did the same with the equivalent fragments of UCBSV as control for comparison. As expected, the

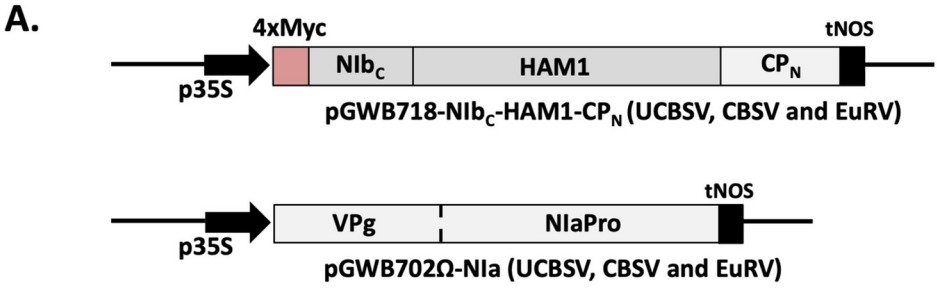

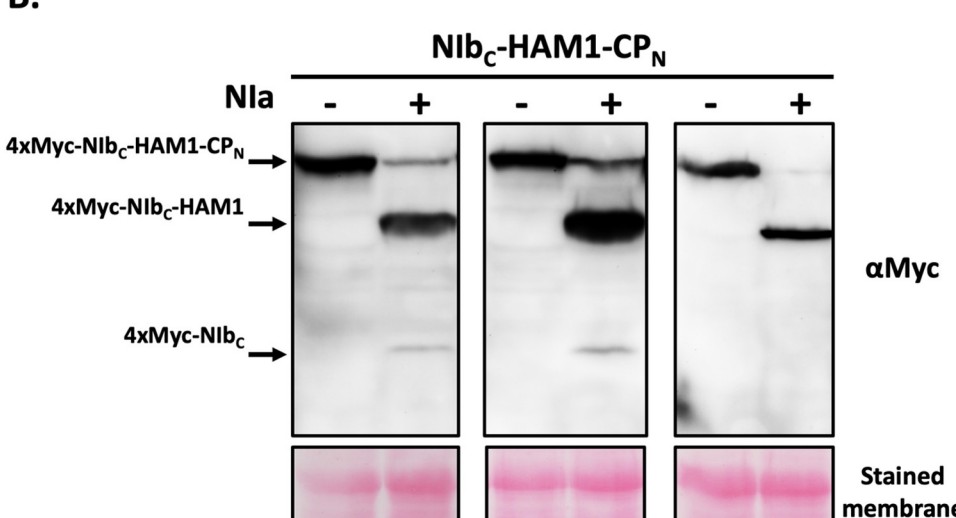

**Fig 8. Suboptimal split of NIb-HAM1 is a general feature of potyvirids.** Schematic representation of constructs based on pGWB702Ω and pGWB718 [30] used for these experiments. p35S: 35S promoter from cauliflower mosaic virus; tNOS: terminator from the NOS gene of *Agrobacterium tumefaciens;* NIb$_C$: NIb C-terminus; CP$_N$: CP N-terminus. (B) Detection of Myc-tagged proteins by immunoblot analysis in samples from *N. benthamiana* leaves expressing NIb$_C$-HAM1-CP$_N$ versions in either the absence (-) or presence (+) of their cognate NIa. Viruses from which the transiently expressed proteins derive are indicated. Blots stained with Ponceau red showing the large subunit of the ribulose-1,5-bisphosphate carboxylase-oxygenase are included at the bottom as a loading control.

expression of UCBSV fragments mimicked the results that we got with the full-length UCBSV-2xMYC virus, so that NIb-HAM1-CP was processed only in the presence of NIa, and it happened at the cleavage site located at the HAM1/CP junction and, with much lower efficiency, at the site placed at the NIb/HAM1 junction (Fig 8B). Remarkably, CBSV behaved just like UCBSV, as the main product, by far, was the one produced after cleavage at the HAM1/CP junction, with just a residual accumulation of the small fragment corresponding to the processing at the NIb/HAM1 junction (Fig 8B). Finally, for EuRV we only detected the product that corresponds to the NIaPro-mediated processing of the cleavage site located between HAM1 and CP (Fig 8B). Altogether, we conclude that most of the NIb and HAM1 might also be covalently bound during CBSV and EuRV infections.

## Discussion

RNA viruses are widespread in nature, where they display a great diversity of particle structures, genome arrangements and expressed proteins [44]. Despite these differences, they are all

replicated by viral-encoded RdRPs sharing, at least in all cases reported so far, a highly conserved core architecture folded into three subdomains (thumb, palm, and fingers) resembling a cupped right hand [45]. In some cases, other key protein domains implicated in viral replication, and/or transcription, are acquired by basic RdRP cores. The flaviviral replicase (NS5), for instance, possesses a capping enzyme domain required to synthetize the 5'-cap structure of genomic RNA [46,47]. The potexviral replicase, in turn, not only has a capping enzyme domain, but also a helicase [48]. Remarkably, the covalent association between a viral RdRP and a HAM1-like protein had not been described so far. Data presented in this study indicate that (i) particular potyvirid replicases are covalently bound to, and work in association with, a HAM1-like pyrophosphatase, and (ii) the requirement of this partnership is host specific, which might be due to the peculiar accumulation of XTP/ITP in some hosts.

Regarding the precise role of viral HAM1 enzymes during the infection, the simple fact that a high fraction of this protein stays covalently attached to the viral replicase strongly suggests that HAM1 participates in replication. As UCBSV- and CBSV-derived HAM1s are pyrophosphatases with preference for non-canonical nucleotides [24], it is logical to hypothesise that HAM1 hydrolyses ITP/XTP in order to prevent their incorporation into the proper RNA polymerase and/or into the viral genome, which would otherwise cause inhibition of RNA synthesis and/or further genome mutations. In other words, it seems quite likely that ITP/XTP behave as natural antiviral molecules, similarly to artificial nucleoside- and nucleotide-like analogues used against plus-stranded RNA viruses in animals [49]. Intriguingly, previous results with CBSV [24] showed that the absence of HAM1 does not increase the complexity of UCBSV and CBSV mutant swarms, thus pointing to a rather direct negative effect of non-canonical nucleotides over the activity of viral RdRPs, as proposed for some synthetic nucleotide analogous in animal virus infections [50].

Theoretically, the concentration of ITP/XTP in the pool of free nucleotides inside cells are tightly maintained at low levels by ITPases to avoid their deleterious effects over DNA and RNA molecules [21]. Therefore, results showing that cassava, and probably other euphorbiaceous, accumulates high amounts of ITP/XTP (Fig 4) question this rule. To conciliate our result in cassava with that broadly accepted idea, we hypothesise that some plants accumulate unexpectedly high concentration of ITP/XTP in certain subcellular compartments, whereas in those locations where they have damaging consequences, such as in the nucleus, ITP/XTP are kept at much lower concentration. The recent suggestion that euphorbiaceous HAM1-like proteins might harbour a nuclear localization signal [51] fits pretty well with this assumption. Therefore, it is possible that viruses infecting plants from the *Euphorbiacea* family (e.g. UCBSV, CBSV, EuRV and CsTLV) have to face high levels of ITP/XTP in the cytoplasm, where they replicate, thus explaining the incorporation of a HAM1 enzyme as an active module of the viral replicase. This possibility also fits well with the expression of some free HAM1 during the infection (Figs 6, 7 and 8), so that it might also help to get rid of ITP/XTP in all those cellular environments where the virus is replicating.

All in all, our findings inform about a novel and interesting case of virus/host coevolution, highlighting (i) the striking peculiarity of cassava plants, and presumably other euphorbiaceous, of accumulating high levels of ITP/XTP into cells, and (ii) the flexibility of RNA viruses to incorporate additional factors when required. Whether this peculiar feature of cassava regarding the high concentration of non-canonical nucleotides evolved as a *bona fide* strategy to prevent multiplication of pathogens, and how this plant copes with the harmful effect of ITP/XTP, are indeed exciting questions deserving special attention in future studies.

## Supporting information

**S1 Fig. Alignment of HAM1 proteins from the indicated potyvirids and organisms.** The intensity of red colour correlates with the degree of amino acid identities. Black asterisk

indicates the fully-conserved K present in all known HAM1 proteins that was mutated by A in this study (see Figs 3 and S2).
(TIF)

**S2 Fig. Measuring the ITPase activity for the wild-type HAM1 and a potential knock-out mutant.** (A) Coupled reaction used to measure ITPase activity. HAM1 ITPase hydrolyses ITP in the presence of Mg+2 to produce IMP and pyrophosphate (PPi). IMP is then oxidized by inosine monophosphate dehydrogenase (IMPDH) in the presence of NAD to produce XMP and NADH2, which is directly monitored in real time at 340 nm. The ITPase activity (U/ml) can be calculated as shown. (B) Cropped polyacrylamide gel stained with coomasie blue that illustrates the yield of MBP-HAM1 in elution fractions. The indicated MBP-HAM1 versions in fraction two, which had the higher protein concentration, are shown. (C) Kinetics of ITP hydrolysis, measured by the formation of NADH2 as indicated in (A), when 1 μl of a 1:10 dilution of the elution fractions shown in (B) were used in the reaction mix.
(TIF)

**S3 Fig. Analysis of a HAM1 partial loss-of-function mutant in different contexts.** (A) Representative pictures under white light of infected and non-treated *N. benthamiana* plants at 11 days post-inoculation. White bar = 4 cm. (B) Representative pictures under white light of upper non-inoculated leaves, at 40 days post-inoculation (dpi), of cassava plants inoculated with the indicated viruses. White arrows indicate the presence of leaf chlorosis induced by UCBSV in cassava. White bar = 4 cm. (C) Representative pictures under white light of stems, at 90 dpi, of cassava plants inoculated with the indicated viruses. Black arrows indicate the presence of brown streaks induced by UCBSV in cassava. (D) Chromatograms of Sanger sequencing results of the DNA fragment of interest amplified by RT-PCR. RNA samples deriving from upper non-inoculated leaves of the indicated cassava plant inoculated with UCBSV--HAM1N$_{35A}$ were used as template. Leaves for RNA preparation were harvested at both 90 and 120 days post-infection (dpi). Coloured circles surround both the original and the naturally introduced mutation. (E) Cropped polyacrylamide gel stained with coomasie blue that shows that equivalent amounts of the indicated proteins were used for the experiment in Fig 4E. 1:20 dilutions (10, 20 and 40 nanograms, correspondingly) were used for the measurement of ITPase activity.
(TIF)

**S4 Fig. Infection symptoms in *Nicotiana benthamiana* produced by 2xMyc-tagged UCBSV mutated in the first amino acid of HAM1.** Representative pictures of infected and non-treated *N. benthamiana plants* taken at 12 days post-inoculation. Bar = 4 cm. This image complements the information shown in Fig 6.
(TIF)

**S5 Fig. Experimental evolution of 2xMyc-tagged UCBSV that carries the T1A mutation in plants multiplied by stem cutting.** Chromatograms of Sanger sequencing results of the DNA fragment of interest amplified by RT-PCR. RNA samples obtained from upper non-inoculated leaves of cassava plants inoculated with the UCBSV-HAM1$_{T1A}$-2xMyc mutant, and from plants derived from a stem cutting passage were used as template. Leaves for RNA preparation were harvested at 60- and 180-days post-infection (dpi). Residues derived from the original mutation and from the spontaneous second mutation are surrounded by a red circle. The amino acid that appears as the reversion to the wild type variant is surrounded by a blue circle.
(TIF)

**S1 Table. Oligonucleotides used in this study.**
(DOCX)

**S2 Table. Templates and name of primers used for PCR amplifications during the construction of the indicated plasmids are shown.**
(DOCX)

**S3 Table. Templates and name of primers used for PCR amplifications during the construction of the indicated plasmids are shown.**
(DOCX)

**S4 Table. Templates and name of primers used for PCR amplifications during the construction of the indicated plasmids are shown.**
(DOCX)

**S5 Table. Relative concentration of the indicated nucleotides in samples from either *Nicotiana benthamiana* or *Manihot esculenta* leaves. For normalization, the average of GTP in *N. benthamiana* is equal to 1.**
(DOCX)

## Acknowledgments

We would like to thank Tsuyoshi Nakagawa, Daniel Silhavy and Gary Foster, for providing Gateway expression vectors, pBIN61-P14 and pYES2-CBSV-F2, respectively. We are also grateful to the Mass Spectrometry Facility (Nucleus, USAL) for its kind help.

## Author Contributions

**Conceptualization:** Adrian A. Valli, Juan Antonio García.

**Funding acquisition:** Adrian A. Valli, Juan Antonio García.

**Investigation:** Adrian A. Valli, Rafael García López, María Ribaya, Francisco Javier Martínez, Diego García Gómez, Beatriz García, Irene Gonzalo, Alfonso Gonzalez de Prádena, Inmaculada Montanuy.

**Resources:** Fabio Pasin.

**Supervision:** Adrian A. Valli, Encarnación Rodríguez-Gonzalo.

**Writing – original draft:** Adrian A. Valli.

**Writing – review & editing:** Adrian A. Valli, Juan Antonio García.

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
