## [Decision Letter · Decision Letter 0]

12 Oct 2021

Dear Dr. Valli,

Thank you very much for submitting your manuscript "Maf/ham1-like pyrophosphatases of non-canonical nucleotides are host-specific partners of viral RNA-dependent RNA polymerases" for consideration at PLOS Pathogens. As with all papers reviewed by the journal, your manuscript was reviewed by members of the editorial board and by several independent reviewers. In light of the reviews (below this email), we would like to invite the resubmission of a significantly-revised version that takes into account the reviewers' comments. Especially the authors need to address the reviewer 1's major concern, i.e., lack of direct evidence of viral inhibition by ITP/XTP. The reviewer 1 also suggested some feasible experiments. We are expecting that the authors can address it timely using these suggested experiments or alternative experiments the authors deem appropriately.

We cannot make any decision about publication until we have seen the revised manuscript and your response to the reviewers' comments. Your revised manuscript is also likely to be sent to reviewers for further evaluation.

Sincerely,

Weifeng Gu

Guest Editor

PLOS Pathogens

Shou-Wei Ding

Section Editor

PLOS Pathogens

Kasturi Haldar

Editor-in-Chief

PLOS Pathogens

orcid.org/0000-0001-5065-158X

Michael Malim

Editor-in-Chief

PLOS Pathogens

orcid.org/0000-0002-7699-2064

Reviewer's Responses to Questions

**Part I - Summary**

Reviewer #1: This manuscript presents a few interesting observations and the authors provided working model, but the conclusions remain largely descriptive without plant host direct genetic evidences. Without evidences based on these mutant plant materials invovled in ITP and XTP biosynthesis, the authors claim a not directly demonstrated conception about UCBSV requires HAM1 to infect Cassava due to elevated levels of non-canonical nucleotides. Second, there is no virus infectivity difference for these UCBSV-HAM1 mutants. Therefore there is no direct relationship between its enzyme activity (HAM1) with virus infection.

Reviewer #2: In this manuscript, using an infectious cDNA clone and reverse genetics approach, the authors found that the Ugandan cassava brown streak virus (UCBSV)-encoding Maf/ham1-like protein is functional when covalently linked to the viral RdRP for adaptation to its natural host plant cassava, in which over-accumulation of non-canonical nucleotides (ITP/XTP) was detected by HPLCMS/MS experiments. This study is novel and provides exciting insights into the evolutionary biology of plant viruses, under the high concentration of non-canonical nucleotides pressure inside the host, by incorporating an ITP/XTP pyrophosphatase into RdRP to support successful replication and infection.

**Part II – Major Issues: Key Experiments Required for Acceptance**

Reviewer #1: 1. In the response letter page 1, the author claimed that "These results are the first ones supporting the idea

that host non-canonical nucleotides play a role in antiviral defense, something considered by other scientists

but not yet published (e.g., https://www.gu.se/om-universitetet/hittaperson/martinlagging

“inhibition of ITPase may point to novel antiviral and antibacterial strategies”). Again the authors claimed that this host-specific constraint is due to an unexpected high concentration of non-canonical nucleotides in cassava but lower ITP and XTP levels in Nicotiana benthamiana. Could you please provide the original set of data for each of 12 samples? Why no statitic analysis for Fig. 4? Also some host genetic manipulation experiments could be provided more direct evidences on this points by silencing some genes invovled in ITP and XTP biosynthesis. Also with these plant genetic manipulation data such as VIGS or CRISPR/CAS9 genome editing will provide a direct evidence to show it is a novel host target for antivirals.

Reviewer #2: The conclusions are appropriate and well supported by the experiments. This reviewer also appreciate that authors have made efforts to address issues in their revision.

**Part III – Minor Issues: Editorial and Data Presentation Modifications**

Reviewer #1: 1. please add Potyvirues before RNA-dependent RNA polymerases in the title.

2. please remove the words such as “Ebola of plants” in the abstract.

3. In the abstract and main text of the manuscript, please consider to replace the word "unexpected high concentration in non-canonical nucleotides in cassava", which is

Reviewer #2: (No Response)
---

## [Decision Letter · Decision Letter 1]

2 Feb 2022

Dear Dr. Valli,

We are pleased to inform you that your manuscript 'Maf/ham1-like pyrophosphatases of non-canonical nucleotides are host-specific partners of viral RNA-dependent RNA polymerases' has been provisionally accepted for publication in PLOS Pathogens.

Best regards,

Weifeng Gu

Guest Editor

PLOS Pathogens

Shou-Wei Ding

Section Editor

PLOS Pathogens

Kasturi Haldar

Editor-in-Chief

PLOS Pathogens

orcid.org/0000-0001-5065-158X

Michael Malim

Editor-in-Chief

PLOS Pathogens

orcid.org/0000-0002-7699-2064

Reviewer Comments (if any, and for reference):

Reviewer's Responses to Questions

**Part I - Summary**

Reviewer #1: The authors have addressed my concerns on the last version of manuscript. No further suggest here.

Reviewer #2: (No Response)

**Part II – Major Issues: Key Experiments Required for Acceptance**

Reviewer #1: N.A.

Reviewer #2: All the concerns have been addressed and discussed. I have no more criticisms.

**Part III – Minor Issues: Editorial and Data Presentation Modifications**

Reviewer #1: N.A.

Reviewer #2: (No Response)

PLOS authors have the option to publish the peer review history of their article (what does this mean?). If published, this will include your full peer review and any attached files.

Reviewer #1: **Yes: **JIAN YE

Reviewer #2: No

---

## [Editor Report · Acceptance letter]

15 Feb 2022

Dear Dr. Valli,

We are delighted to inform you that your manuscript, "Maf/ham1-like pyrophosphatases of non-canonical nucleotides are host-specific partners of viral RNA-dependent RNA polymerases," has been formally accepted for publication in PLOS Pathogens.

Best regards,

Kasturi Haldar

Editor-in-Chief

PLOS Pathogens

orcid.org/0000-0001-5065-158X

Michael Malim

Editor-in-Chief

PLOS Pathogens

orcid.org/0000-0002-7699-2064